

# Dynamics of locomotion in the seed harvesting ant *Messor barbarus:* effect of individual body mass and transported load mass

Hugo Merienne, Gérard Latil, Pierre Moretto and Vincent Fourcassié

Centre de Recherches sur la Cognition Animale, Centre de Biologie Intégrative, Université de Toulouse, CNRS, UPS, Toulouse, France

## ABSTRACT

Ants are well-known for their amazing load carriage performances. Yet, the biomechanics of locomotion during load transport in these insects has so far been poorly investigated. Here, we present a study of the biomechanics of unloaded and loaded locomotion in the polymorphic seed-harvesting ant *Messor barbarus* (Linnaeus, 1767). This species is characterized by a strong intra-colonial size polymorphism with allometric relationships between the different body parts of the workers. In particular, big ants have much larger heads relative to their size than small ants. Their center of mass is thus shifted forward and even more so when they are carrying a load in their mandibles. We investigated the dynamics of the ant center of mass during unloaded and loaded locomotion. We found that during both unloaded and loaded locomotion, the kinetic energy and gravitational potential energy of the ant center of mass are in phase, which is in agreement with what has been described by other authors as a grounded-running gait. During unloaded locomotion, small and big ants do not display the same posture. However, they expend the same amount of mechanical energy to raise and accelerate their center of mass per unit of distance and per unit of body mass. While carrying a load, compared to the unloaded situation, ants seem to modify their locomotion gradually with increasing load mass. Therefore, loaded and unloaded locomotion do not involve discrete types of gait. Moreover, small ants carrying small loads expend less mechanical energy per unit of distance and per unit of body mass and their locomotion thus seem more mechanically efficient.

## INTRODUCTION

Locomotion is a crucial aspect of animal behavior. It is essential to accomplish tasks such as searching for food or a shelter, hunting for prey, looking for a mate or escaping a predator. For each of these tasks, animals have to adjust specific features of their locomotion in order to behave optimally (*Halsey, 2016*). Different ways of moving are thus used by animals, each most fitted to a given situation. Among walking animals, insects are of particular interest for the study of locomotion due to their outstanding performances, as attested by the maximum speed some insects can reach, e.g., about 40 body lengths per

Corresponding author
Vincent Fourcassié,
vincent.fourcassie@univ-tlse3.fr

second for the ant *Cataglyphis bombycina* (*Pfeffer et al., 2019*) or about 35 body lengths per second for the cockroach *Periplaneta americana* (*Full & Tu, 1991*). This probably explains why insects have been for decades a source of inspiration for the design of legged robots (*Kar, Kurien Issac & Jayarajan, 2003*; *Koditschek, Full & Buehler, 2004*; *Dupeyroux, Serres & Viollet, 2019*).

From a purely kinematic point of view, the most common locomotory gait encountered in insects is the alternating tripod gait (*Delcomyn, 1981*), in which the swing phase of a set of three legs called tripods (the ipsilateral front and hind leg and the contralateral mid leg) is synchronized with the contact phase of the contralateral tripod. However, this pattern can be altered by many factors. For example, it can vary with the speed (*Bender et al., 2011*; *Wosnitza et al., 2012*; *Mendes et al., 2013*; *Wahl, Pfeffer & Wittlinger, 2015*), the behavior (exploration: *Reinhardt, Weihmann & Blickhan, 2009*; *Reinhardt & Blickhan, 2014*; wall-following: *Bender et al., 2011*; backward locomotion: *Pfeffer, Wahl & Wittlinger, 2016*, the external (leg amputation: *Fleming & Bateman, 2007*; *Gruhn, Zehl & Büschges, 2009*; *Grabowska et al., 2012*) and internal state (effects of ageing: *Ridgel & Ritzmann, 2005*); blocking of proprioceptive feedback : *Mendes et al., 2013*) of the insects, as well as with the characteristics of their physical environment, such as the type of substrate on which they walk (*Spence et al., 2010*), the presence of wind (*Full & Koehl, 1993*), the slope of the terrain (*Diederich, 2006*; *Seidl & Wehner, 2008*; *Moll, Roces & Federle, 2010*; *Grabowska et al., 2012*; *Wöhrl, Reinhardt & Blickhan, 2017*), and the presence of obstacles (*Watson et al., 2002*).

One of the factors that is known to affect locomotory gait in humans (*Ahmad & Barbosa, 2019*) and other vertebrates (review by *Jagnandan & Higham, 2018*), but that has so far received little attention in insects, is load carriage. Load carriage occurs in insects mostly internally, for example after ingesting food or when a female insect carry eggs. However, these internal loads only induce small changes in the total mass of individuals. Much more impressive are the external loads that are carried by some insects while returning to their nest. In ants in particular, these loads can be considerable and weigh more than ten times the body mass of individuals (*Bernadou et al., 2016*). They can shift the center of mass (CoM) of individuals forward and thus have a strong impact on their locomotion. The changes induced by load carriage on the locomotion of ants have so far been investigated only with a kinematic approach, through the analysis of stepping pattern (*Zolliköfer, 1994*; *Moll, Roces & Federle, 2013*; *Merienne et al., 2020*). In the seed harvesting ant *Messor barbarus* for example, load carriage has been found to decrease locomotory speed (through a decrease in stride frequency but not of step amplitude), to increase the mean number of legs in contact with the ground, as well as to induce a change in leg positioning, with ants spreading their legs further away from their longitudinal body axis in order to maintain their stability (*Merienne et al., 2020*). On the other hand, the impact of load carriage on the exchanges of mechanical energies and on the mechanical cost of locomotion in ants is poorly documented. Here, we aim to fill this gap by investigating the impact of load carriage on the CoM dynamics in individuals of the species *M. barbarus* (Linnaeus, 1767) whose workers routinely transport items weighing up to thirteen times their own mass over dozen of meters (*Bernadou et al., 2016*). Individuals of this species show a high variation

in size within colonies, with a body mass ranging from 1.7 to 40.0 mg. This variation is continuous and is characterized by a positive allometry between head size and thorax length (*Heredia & Detrain, 2000*; *Bernadou et al., 2016*), which means that, relative to their size, the head of large workers is bigger than that of small workers. Consequently, the CoM of big workers is shifted forward compared to that of small workers (*Bernadou et al., 2016*; see also *Anderson, Rivera & Suarez, 2020* for ants of the genus *Pheidole*). In our study we thus chose to investigate both the effect of body mass and load mass on the locomotion of loaded ants. We varied in a systematic way the mass of the load carried by ants of different sizes so as to cover the same range of load ratio. We then compared the displacement of the CoM and its mechanical work, which represents the amount of mechanical energy needed to raise the CoM and accelerate it during locomotion, of the same individuals in unloaded and loaded condition. Since external load carriage is already observed in wasps (*Polidori et al., 2013*), which are considered as the ant ancestors (*Peters et al., 2017*), we hypothesized that ants could have evolved some mechanisms to transport loads economically. Specifically, we tested the assumption that, ants, in the same way as humans (*Heglund et al., 1995*), could be able to decrease, or at least compensate, the additional mechanical cost of carrying a load by improving the pendulum-like behavior of their CoM through a better transfer between the gravitational potential and kinetic energy of their CoM. Moreover, since large ants have a less stable locomotion than small ants (*Merienne et al., 2020*) due to the forward shift of their CoM, we predict that their locomotion when transporting loads representing the same amount of individual body mass should be less mechanically efficient than that of small ants, and the more so for loads of increasing mass.

## MATERIAL AND METHODS

Note that the data presented in this paper are part of the data collected in the study presented in *Merienne et al. (2020)*. The studied species, experimental setup and experimental protocol are thus the same.

### Studied species

Experiments were carried out with a large colony of *M. barbarus* collected in April 2018 at St Hippolyte (Pyrénées Orientales), on the French Mediterranean coast. Workers in the colony ranged from 2 to 15 mm in length and from 1 to 40 mg in body mass. The colony was housed in glass tubes with a water reservoir at one end and kept in a room at 26 °C with a 12:12 L:D regime. The tubes were placed in a box (LxWxH: $0.50 \times 0.30 \times 0.15$ m) whose walls were coated with Fluon® to prevent ants from escaping. During the experimental period, ants were fed with a mixture of seeds of various species and had access *ad libitum* to water.

### Experimental setup

Ants were tested on a setup designed and built by a private company (R&D Vision, France. http://www.rd-vision.com/). It consisted in a walking platform surrounded by five high speed cameras (JAI GO-5000M-PMCL: frequency: 250 Hz; resolution: 30 µm/px for the top camera, 20 µm/px for the others). One camera was placed above the platform and four

were placed on its sides. The platform was 160 mm long and 25 mm wide and was covered with a piece of black paper (Canson®, 160 g/m²). Four infrared spots ($\lambda = 850$ nm, pulse frequency= 250 Hz) synchronized with the cameras illuminated the scene from above. The mean temperature in the middle of the platform, measured with an infrared thermometer (MS pro, Optris, USA, http://www.optris.com) over the course of the experiment, was (mean ± SD) 28 ± 1.4 °C.

## Experimental protocol

We performed all experiments between April and July 2018.

We wanted to make sure that the ants we tested were foraging workers. Therefore, the first day of an experimental session, we selected a random sample of workers returning to their nest with a seed on a foraging trail established between the box containing the colony and a seed patch. We then kept these ants in a separate box and used them in our experiments the following days.

Each ant was tested twice: the first time unloaded and the second time loaded with a fishing lead glued on its mandibles. Before being tested, unloaded ants were first weighed to the nearest 0.1 mg with a precision balance (NewClassic MS semi-micro, Mettler Toledo, United States). Individual ants were then gently placed at one end of the platform and we started recording their locomotion as soon as they entered the camera fields. The recording was retained only if ants walked straight for at least three full strides, a stride being defined as the interval of time elapsed between two consecutive lift off of the right mid leg. All videos were subsequently cropped to a whole number of strides. To stimulate the ants and to obtain a straighter path, an artificial pheromone trail was laid down along the middle axis of the platform by depositing every centimeter a small drop of a hexane solution of Dufour gland (1 gland/20µl), which is responsible for the production of trail pheromone in *M. barbarus* (*Heredia & Detrain, 2000*). This operation was renewed every 45 min in order to keep a fresh trail on the platform.

Once five ants were tested in unloaded condition, we proceeded with the test in loaded condition. First, each ant was anesthetized by putting it in a vial plunged in crushed ice. It was then fixed on its back, with its head maintained horizontally, and we glued a calibrated fishing lead on its mandibles with a droplet of superglue (Loctite, http://www.loctite.fr/). After letting the glue dry for 15 min and the ant recover for half an hour, the ant was placed again on the platform and its locomotion was recorded in loaded condition. We retained only the recordings in which the load did not touch the ground during the transport (see *Merienne et al., 2020*). At the end of the recording, the ant was captured and weighed a second time. It was then killed and each of its body parts (head, thorax, gaster) was weighed separately.

## Data extraction and analysis

In order to compute the 3D displacement of the ants' main body parts (head, thorax, gaster) and of its overall CoM, we tracked several anatomic points on the view of the top camera (Figs. 1A–1C) and on the view of one of the side cameras (Figs. 1B–1D) with the software Kinovea (version 0.8.15, https://www.kinovea.org).
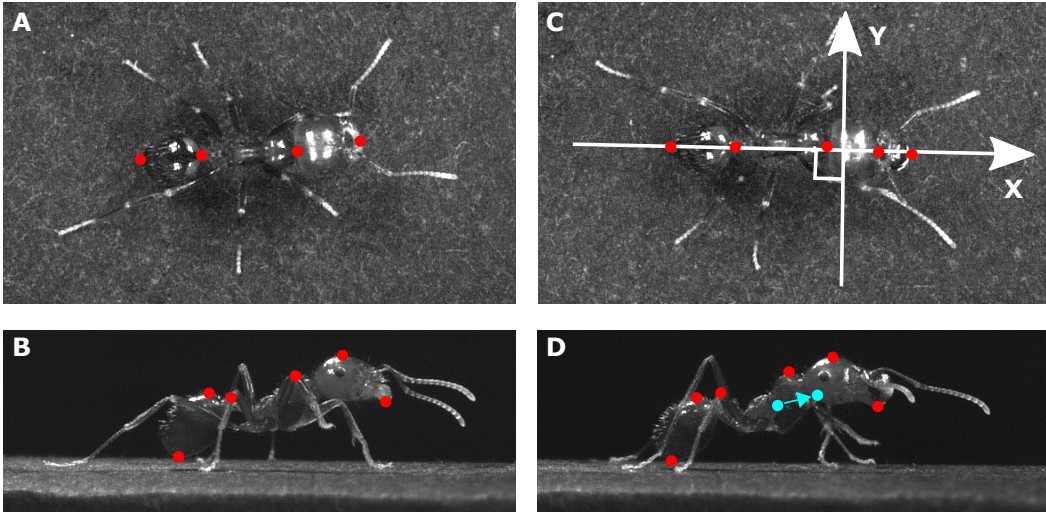

**Figure 1** **Position of the tracked points on each ant.** The pictures show (A, C) a view from the top and (B, D) a view from the side of the same ant (mass = 10.1 mg) tested in (A, B) unloaded and (C, D) loaded condition (load mass = 3.5mg). The $X$ axis in (C) stands for the longitudinal body axis while the $Y$ axis stands for the transverse body axis. The tracked points are shown in red. The filled blue points in (D) show the positions of the overall CoM of the ant in the unloaded and loaded condition. The arrow shows the shift in the position of the overall CoM between the unloaded and loaded condition.

We assumed a homogeneous distribution of the mass within each body parts and thus computed the $(X, Y)$ coordinates of the CoM of the three main body parts (plus the load) as the mean of the $(X, Y)$ coordinates of the two points tracked at their extremities on the top view and the vertical position $(Z)$ as the mean of the vertical position of the two points tracked on each of these parts on the side view. For each frame we computed the position of the overall CoM of an ant as the barycenter of the CoM of its three main body parts (plus the load for loaded ants) weighted by their mass. For each ant tested, we delimited the different strides on the videos and then, for each stride, we calculated the positions $(X, Y, Z)$ and velocity vectors of the overall CoM. Finally, we averaged the CoM speeds and positions across the multiple stride cycles in order to obtain a single mean trajectory of the CoM in each condition (unloaded and loaded).

In order to characterize the mean trajectories of the CoM for each ant and condition, we computed the peak-to-peak amplitude of the $Z$ positions of the CoM and assessed the sinus-like behavior of the changes in $Z$ position and in the norm of the velocity vector. In order to do so, we first normalized the $Z$ positions and the values of the norm of the velocity vector by their respective peak-to-peak amplitude and fitted a sinus function to the resulting signals. We then computed the root-mean-square error (RMSE) between the fitted function and the normalized data.

In order to assess the general posture of the ants during locomotion, we also computed the mean $Z$ position of their CoM in units of body length and the mean inclination angle of their body during locomotion (defined as the angle between the horizontal $X$ axis and the line linking the gaster and head CoMs).

From the dynamic of the CoM, we then computed its kinetic $E_k$ and gravitational potential $E_p$ energies relative to the surroundings with the formulae

$$E_k = 0.5 * m * v^2 \tag{1}$$

and

$$E_p = m * g * h \tag{2}$$

where $m$ is the mass of the ant (plus the mass of the load if one is carried), $v$ the speed of the CoM, $g$ the gravitational constant and $h$ the vertical position of the CoM above the walking platform. We then computed the external mechanical energy of the CoM as the sum of the kinetic and potential energies.

Finally, following *Bastien et al. (2016)*, we computed the external mechanical work ($W_{ext}$) achieved to raise and accelerate the CoM as the sum of the positive increments of the external mechanical energy. Since ants did not walk the same distance or during the same amount of time, in order to compare the mechanical work they achieved, we divided $W_{ext}$ by the distance travelled and thus obtained a "mechanical work per unit distance" $W_{ext,d}$. This makes sense if one considers that locomotion is a repetitive process and that we cropped our videos to a whole number of strides. We then computed the mean external power ($P_{ext}$) by dividing $W_{ext}$ by the duration of locomotion. Finally, we computed the mass specific values of $W_{ext,d}$ and $P_{ext}$ by dividing both of these metrics by the ant mass for unloaded locomotion and the ant mass plus load mass for loaded locomotion.

Following *Cavagna, Thys & Zamboni (1976)* we then computed the energy recovered ($R$, expressed in percentage) through the pendulum-like oscillations of the CoM with the formula:

$$R = 100 * \frac{W_k + W_p - W_{ext}}{W_k + W_p} \tag{3}$$

where $W_k$ is the sum of the positive increments of the kinetic energy versus time curve and $W_p$ is the sum of the positive increments of the potential energy versus time curve. $R$ is an indicator of the amount of energy transferred between the potential and the kinetic energy of the CoM due to its pendulum-like behavior: the closer the value of $R$ to 100%, the more consistent the locomotor pattern is with the Inverted Pendulum System (IPS) model (*Cavagna, Heglund & Taylor, 1977*) in which the fluctuations of $E_p$ and $E_k$ are perfectly out of phase, i.e., all the kinetic energy of the CoM is transformed in potential energy, and vice versa, over a stride.

In order to further characterize the relationship between $E_k$ and $E_p$, we computed the Pearson correlation coefficient between $E_k$ and $E_p$, and, following *Ahn, Furrow & Biewener (2004)* and *Vereecke, D'Août & Aerts (2006)*, the percentage congruity between $E_k$ and $E_p$ (defined as the percentage of time $E_k$ and $E_p$ changed in the same direction). We then fitted a sinus function to the variations of both $E_k$ and $E_p$, extracted the phase of $E_k$ and $E_p$ from these sinus functions, and computed the difference between the two phases in order to access the phase lag between $E_k$ and $E_p$ (a positive value of this lag indicating that $E_k$ is late compared to $E_p$).

For the unloaded condition, we expressed all variables $Y$ as a power law function of ant mass $M$, i.e., $Y = a * M^b$ (*Merienne et al., 2020*). For each variable, the values of the coefficients $a$ and $b$, as well as the value of the variable predicted by the statistical model for the mean mass of the tested ants (12.5 mg), are given in a table, along with their 95% confidence interval. For the loaded condition, we computed for each ant the ratio of the value of each variable between the loaded $Y_l$ and unloaded $Y_u$ condition. This ratio was then expressed as a power law function of both ant mass $M$ and load ratio $LR$, defined as 1 + (load mass/ant body mass) (*Bartholomew, Lighton & Feener Jr, 1988*), i.e., $\frac{Y_l}{Y_u} = c * M^d * LR^e$ (*Merienne et al., 2020*). The value of the coefficients $c$, $d$ for ant mass, $e$ for load ratio for each variable, as well as the value of the variable predicted by the statistical model for the mean mass of tested ants and a load ratio of one, along with their 95% confidence interval, are given in a table. The coefficients $d$ and $e$ are positive when the response variable increases with increasing value of ant mass and load ratio, they are negative in the other case.

All data analyses were performed and graphics designed with R (v. 3.5.1) run under RStudio (v. 1.0.136). The *confint()* function was used to calculate the confidence intervals of the model coefficients.

## RESULTS

In total, 52 ants whose body mass ranged from 1.7 to 33.0 mg were tested in both unloaded and loaded conditions, with a load ratio ranging from 1.2 to 7.0 (Fig. 2).

### Unloaded ants: influence of body mass

The analysis of the position of the CoM shows that there was no evidence of a periodic pattern on the Y axis. On the Z axis on the other hand, the position of the CoM (Fig. 3A), as well as its speed norm (Fig. 3B), followed a periodic pattern that was well approximated by a sinus function, as shown by the low value of the RMSE (Table 1, line 1 & 2). Interestingly, the amplitude of the oscillations of the CoM Z position seems to be approximately the same for small and big ants (Fig. 3A). Indeed, the relative amplitude (expressed in units of body length, Table 1, line 3) of the oscillations of the CoM Z position, as well as its mean relative position (Table 1, line 4), decreased significantly with increasing ant mass ($F_{1,52} = 75.88$, $P < 0.001$ and $F_{1,52} = 105.24$, $P < 0.001$, respectively). The CoM of big ants was thus relatively lower and oscillated with a relatively smaller amplitude than that of small ants. The ant body angle was independent of ant mass (Table 1, line 5).

The variations of $E_k$ and $E_p$ were periodic and the amplitude of $E_p$ was much greater than that of $E_k$ in both small (Fig. 4A) and big ants (Fig. 4B). $E_k$ and $E_p$ were mostly in phase, as shown by the high values of both the correlation coefficient (Table 1, line 6) and the percentage congruity (Table 1, line 7). Nevertheless, $E_k$ and $E_p$ were more in phase for small ants than for big ants (Fig. 5A). The phase lag between the variation of potential and kinetic energies was positive (Fig. 5B) and increased with increasing ant mass (Table 1, line 8: $F_{1,52} = 11.51$, $P = 0.001$). As a consequence, $E_k$ and $E_p$ were more out of phase for big ants compared to small ants and thus both the correlation coefficient (Table 1, line 6) and

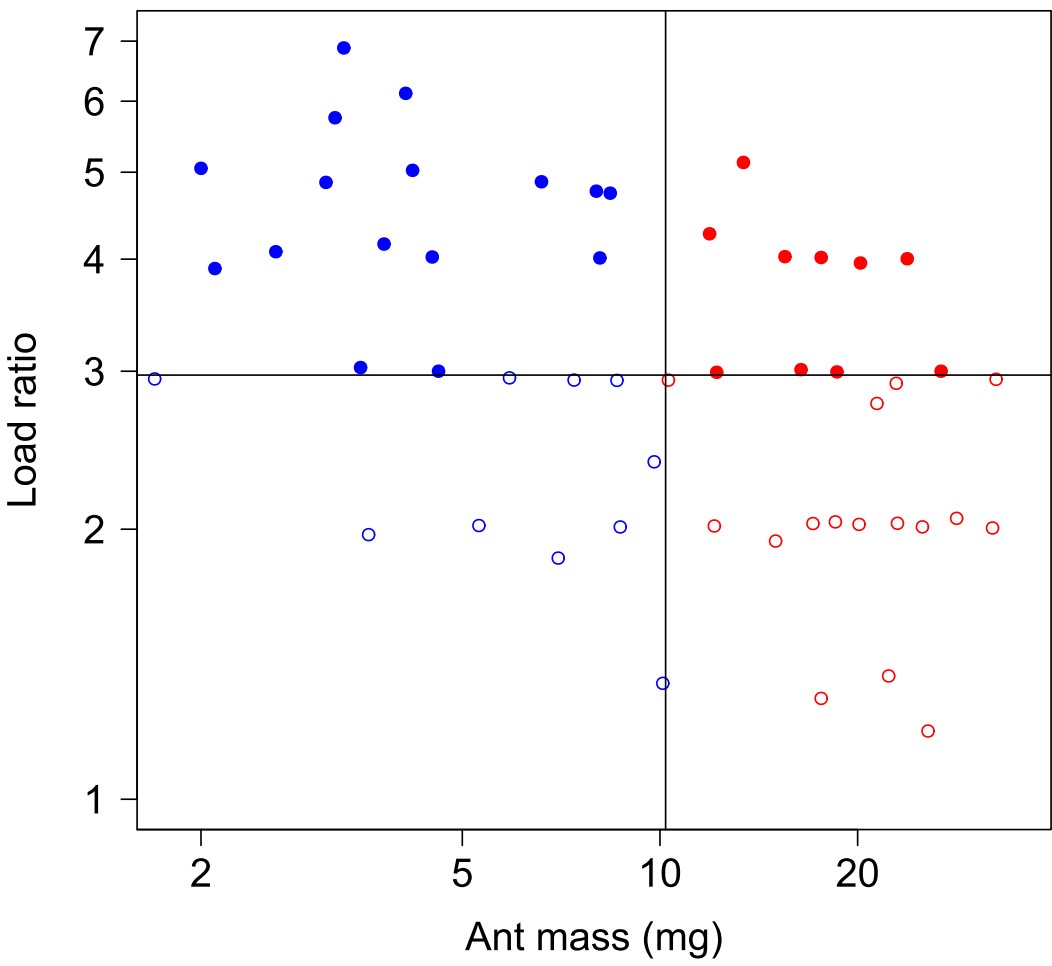

**Figure 2 Body mass and load ratio of tested ants.** The points represent small ants (blue, $N = 27$), big ants (red, $N = 27$), low load ratio (empty dots, $N = 27$) and high load ratio (filled dots, $N = 27$). The thin vertical and horizontal lines correspond to the median body mass and median load ratio, respectively.

the percentage congruity (Table 1, line 7) decreased with increasing ant mass ($F_{1,52} = 5.79$, $P = 0.020$ and $F_{1,52} = 4.75$, $P = 0.034$, respectively).

The external mechanical work of the CoM per unit distance ($W_{ext,d}$) increased with increasing ant mass (Fig. 6A). However, there was no relationship between the mass-specific external mechanical work of the CoM per unit distance ($W_{ext,d}/m$) and ant mass ($m$) (Table 1, line 9). In the same way, the mean external mechanical power of the CoM ($P_{ext}$) increased with increasing ant mass (Fig. 6B) but there was no relationship between the mass-specific external mechanical power of the CoM ($P_{ext}/m$) and ant mass (Table 1, line 10).

The percentage of energy recovery was very low and did not depend on ant mass (Table 1, line 11).

### Loaded ants: influence of ant mass and load ratio

In the same way as in unloaded condition, no periodicity was found in the CoM $Y$ trajectory in the loaded condition. On the $Z$ axis, independent of ant mass, the sinus-like periodicity

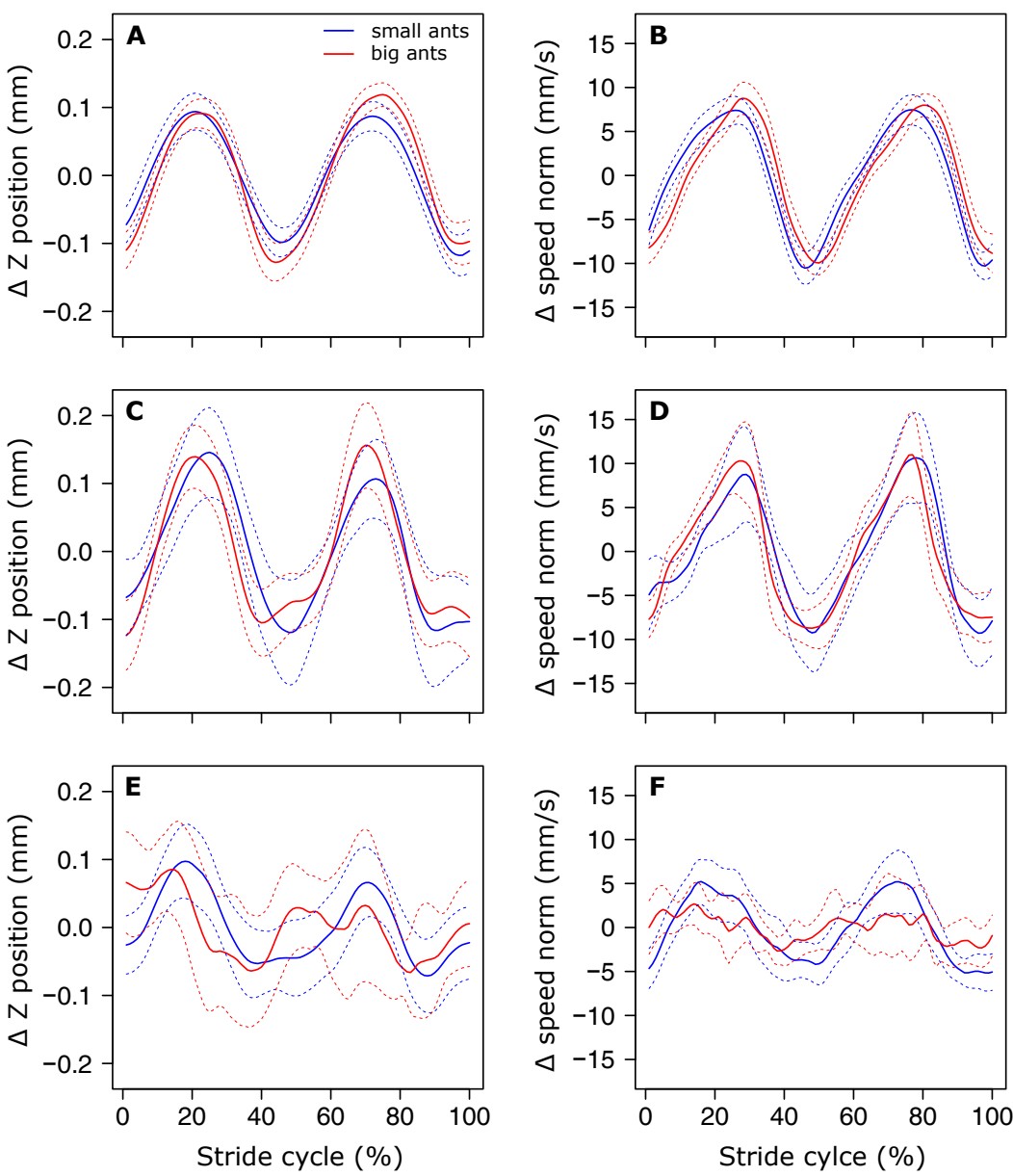

**Figure 3** **Variation of the vertical position and norm of the velocity vector of the ant overall CoM.** (A, C, E) Mean variation of the vertical position and (B, D, F) norm of the velocity vector of the CoM. (A, B) small (blue, ant mass < 10.2 mg, $N = 27$) and big (red, ant mass > 10.2 mg, $N = 27$) for unloaded ants over one stride cycle. small (blue, ant mass < 10.2 mg, $N = 9$) and big (red, ant mass > 10.2 mg, $N = 18$) ants loaded with small load ratio (LR< 3). (E-F) The dashed lines represent the 95% confidence interval of the mean. For the sake of clarity, all values are centered on their mean.

of the $Z$ position of the CoM (assessed by the $Z$ position RMSE) decreased with increasing load ratio (Figs. 3C and 3E, Table 2, line2: $F_{1,52} = 3.87$, $P = 0.010$). We found no significant changes in the relative amplitude of the oscillations of the CoM $Z$ position (Table 2, line 3) and in the mean $Z$ position of the CoM (Table 2, line 4) between the unloaded and

**Table 1 Effect of body mass on the kinematics of unloaded ants.** The results of a power law model describing the influence of ant mass M (in mg) on each variable $Y$, with $Y = a$ Mb, are indicated on each line of the table. The first column gives the model prediction, along with its 95% confidence interval, for the mean value of ant masses (12.5 mg). The second and third column give the value of the coefficient a and b for ant mass respectively, along with their 95% confidence interval. The adjusted R for the model is given in the fourth column. Bold characters indicate that 0 is not included in the 95% confidence interval of the coefficient b for ant mass. $N = 52$ ants.

| | Variable | Model prediction for mean (ant mass) [CI] | Coefficient a [CI] | Coefficient b for ant mass [CI] | Adj $R^2$ |
|---|---|---|---|---|---|
| 1 | RMSE speed norm | 0.134 [0.124;0.145] | 0.148 [0.119;0.184] | $-0.038$ [$-0.129$; 0.052] | 0.00 |
| 2 | RMSE Z position | 0.143 [0.129;0.158] | 0.160 [0.121;0.212] | $-0.044$ [$-0.161$; 0.073] | 0.00 |
| 3 | Z position amplitude (BL[1]) | 0.015 [0.014;0.017] | 0.048 [0.037;0.062] | **$-0.451$ [$-0.555$;$-0.347$]** | 0.59 |
| 4 | Mean Z position (BL) | 0.121 [0.115;0.128] | 0.278 [0.238;0.324] | **$-0.326$ [$-0.389$;$-0.262$]** | 0.67 |
| 5 | Body angle (°) | 11.77 [10.85;12.76] | 14.71 [11.68;18.52] | $-0.088$ [$-0.183$; 0.008] | 0.04 |
| 6 | Correlation coefficient | 0.411 [0.355;0.475] | 0.695 [0.459;1.053] | **$-0.206$ [$-0.379$;$-0.034$]** | 0.09 |
| 7 | Percentage congruity (%) | 66.18 [64.33;68.09] | 72.62 [66.97;78.74] | **$-0.036$ [$-0.070$;$-0.003$]** | 0.07 |
| 8 | Ek / Ep phase (°) | 26.42 [21.81;32.00] | 9.864 [5.637;17.26] | **0.387 [ 0.157; 0.616]** | 0.18 |
| 9 | Mass specific Wext (nJ/mm/mg) | 1.072 [1.027;1.120] | 1.050 [0.929;1.187] | 0.008 [$-0.043$; 0.059] | 0.00 |
| 10 | Mass specific Pext (nJ/s/mg) | 30.94 [28.58;33.49] | 29.32 [23.40;36.75] | 0.021 [$-0.073$; 0.115] | 0.00 |
| 11 | Percentage recovery (%) | 8.200 [7.392;9.097] | 6.407 [4.770;8.606] | 0.097 [$-0.026$; 0.219] | 0.03 |

**Notes.**

BL, Body Length

loaded condition, whatever the ant mass and load ratio. The speed of the CoM in loaded condition followed a periodic pattern (Figs. 3D and 3F) that was well approximated by a sinus function, whatever the values of ant mass and load ratio (Table 2, line 1). Independent of ant mass and load ratio, the ant body angle did not change between the unloaded and loaded condition (Table 2, line 5).

In the same way as in unloaded condition, $E_k$ and $E_p$ were mostly in phase for low load ratio in small (Fig. 4C) and big ants (Fig. 4D), but less so for high load ratio (Figs. 4E and 4F). Independent of ant mass and load ratio, the correlation coefficient between $E_k$ and $E_p$ did not vary significantly between the unloaded and loaded condition (Fig. 5A, Table 2, line 6) and the phase lag only slightly decreased (Fig. 5B, Table 2, line 8). However, independent of ant mass, the percentage congruity decreased for ants carrying loads of increasing load ratio (Table 2, line 7: $F_{1,52} = 8.22$, $P<0.001$). In the loaded condition, in the same way as in the unloaded condition, $E_k$ and $E_p$ were more in phase for small ants than for big ants (Fig. 5A). However, contrary to the unloaded condition, the phase lag was not statistically different between small and big ants in the loaded condition (Fig. 5B).

Independent of load ratio, the mass-specific $W_{ext,d}$ increased with increasing ant mass (Table 2, line 9: $F_{2,51} = 12.47$, $P = 0.024$) and, independent of ant mass, it also increased with increasing load ratio ($F_{2,51} = 12.47$, $P<0.001$). However, there was no effect of the load on the mass-specific $P_{ext}$ (Table 2, line 10). Finally, there was no significant change in percentage recovery between the unloaded and loaded condition (Table 2, line 11).

## DISCUSSION

In this study, we investigated the dynamics of locomotion of unloaded and loaded individuals of the polymorphic ant *M. barbarus*. We found that during unloaded

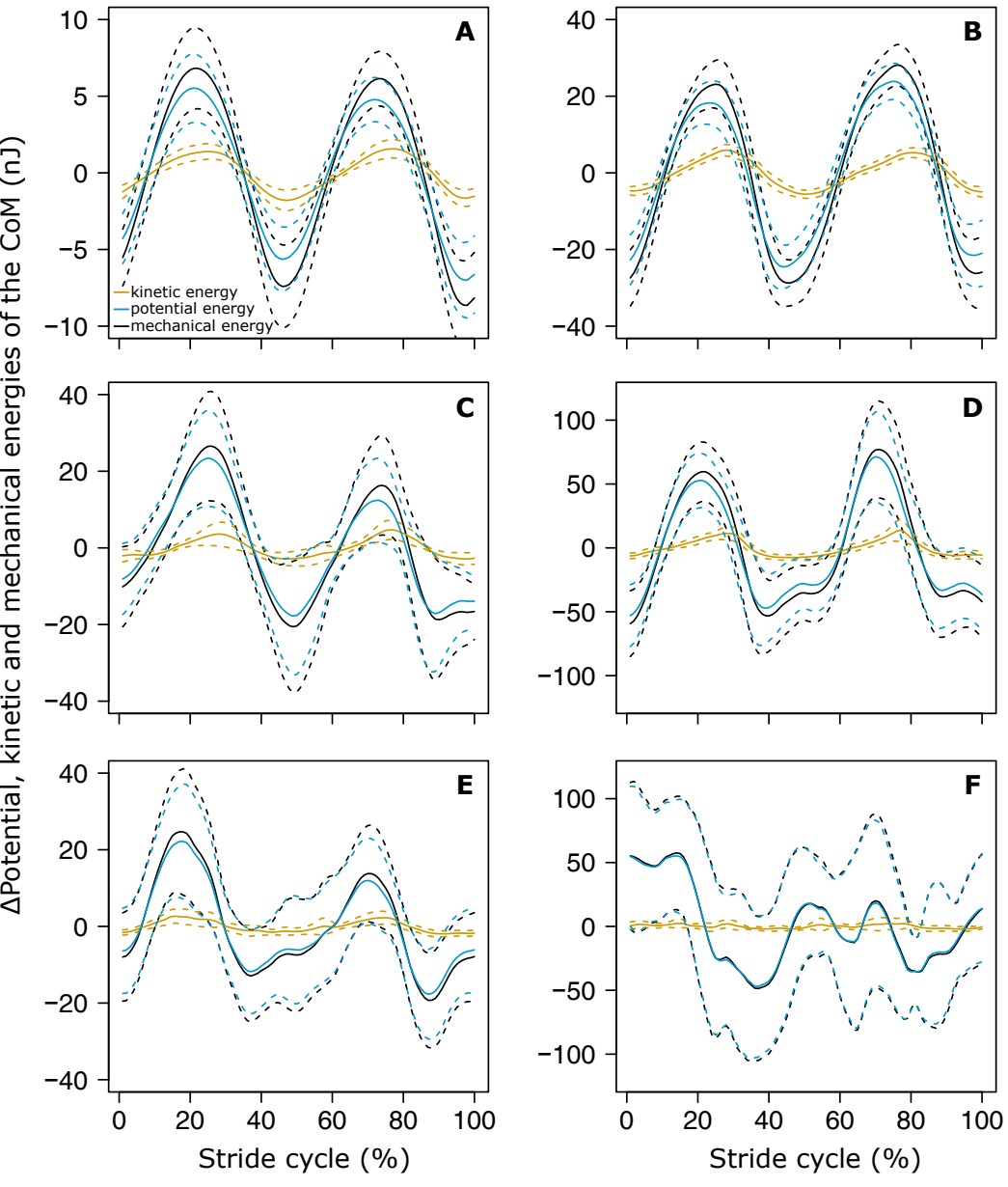

**Figure 4** **Variation of the mechanical energies of the CoM relative to the surroundings.** The mean variation of the kinetic (orange), potential (light blue) and external (black) mechanical energies over one stride cycle are shown for (A) small unloaded ants (ant mass < 10.2 mg, $N = 27$). (B) Big unloaded ants (ant mass > 10.2 mg, $N = 27$). (C) Small loaded ants with small load ratio (ant mass < 10.2 mg, load ratio < 3, $N = 9$). (D) Big loaded ants with small load ratio (ant mass > 10.2 mg, load ratio < 3, $N = 17$). (E) Small loaded ants with high load ratio (ant mass < 10.2 mg, load ratio > 3, $N = 18$). (F) Big loaded ants with high load ratio (ant mass ¿10.2 mg, load ratio > 3, $N = 10$). For the sake of clarity, the values of energies are centered on their mean.

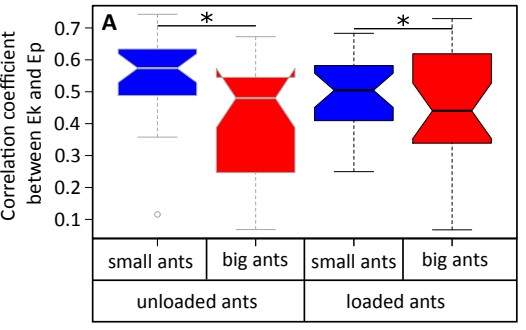
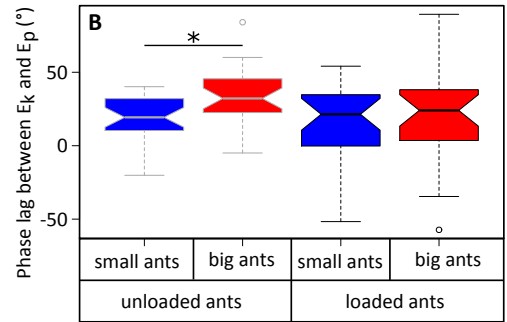

**Figure 5** **Correlation coefficient and phase lag between the kinetic and potential energies of the CoM.**
(A) Correlation coefficient and (B) phase lag between the CoM $E_p$ and $E_k$ for unladen ants and loaded ants. The results are shown for small (blue) and big ants (red). * indicates that the difference between samples is significant according to a Welch two sample $t$-test ($P < 0.05$). The line within the box represents the median, the lower and upper boundaries represent respectively the 25th and 75th percentiles while the whiskers extend to the smallest and largest values within 1.5 box lengths. The notch in each bar represents the confidence interval of the median. $N = 52$ ants.

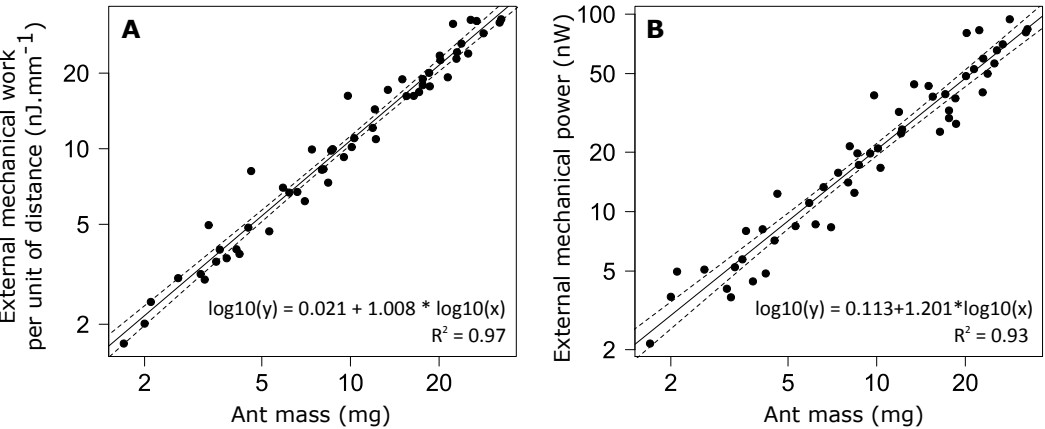

**Figure 6** **External mechanical work and power for unloaded ants.** (A) External mechanical work ($F_{1,52} = 1502$, $P < 0.001$) and (B) external mechanical power ($F_{1,52} = 717$, $P < 0.001$). The straight line gives the prediction of a linear regression model and the dashed lines the 95% confidence interval of the slope of the regression line ($N = 52$ ants).

locomotion the variations of the speed of the CoM and of its vertical position are characterized by a periodic pattern, with two periods corresponding to the two steps included in one stride. These variations were well described by a sinus function, although the pattern of variation of the CoM Z position was strongly affected by load transport. The kinetic and potential energies were mostly in phase during unloaded locomotion, which led to very low energy recovery values. With increasing load however, the variations in potential energy became much greater than the variations in kinetic energy. Therefore, ants achieved mechanical work mainly to raise their CoM rather than to accelerate it. The external mechanical work ants had to perform to raise and accelerate their CoM over a

locomotory cycle did not vary with body mass for unloaded ants and increased with load ratio for ants of same body mass.

## Unloaded ants

During unloaded locomotion, the mean of the absolute $Z$ position of the CoM, as well as the amplitude of its variations, did not differ between small and big ants. Therefore, relative to their size, the body of small ants was higher over the ground than that of big ants and their CoM made greater vertical oscillations. This difference cannot be explained by a change in body inclination because this latter did not change between small and big ants. It thus seems that small ants are walking in a more erect posture than big ants. This could be related to a more excited state of small ants compared to big ants in response to manipulation, as also suggested by their higher locomotory speed relative to their size (*Merienne et al., 2020*). Such a difference between ants of different sizes in response to threat has already been found in other ant species, e.g., the leaf-cutting ant *Atta capiguara* (*Hughes & Goulson, 2001*), and this could be related to the division of labor within colonies. Further experiments should be performed to answer this question.

The kinetic and potential energies of the CoM were mainly in phase during unloaded locomotion, which led to very low energy recovery values (7–9%). These values are similar to those reported by *Full & Tu (1991)* in the cockroach *Periplaneta americana* and a bit below those reported in the cockroach *Blaberus discoidalis* (*Full & Tu, 1990*) and in the ant *Formica polyctena* (*Reinhardt & Blickhan, 2014*). These values are not consistent with the inverted pendulum model of *Cavagna, Heglund & Taylor (1977)*. As walking ants never display aerial phases (*Merienne et al., 2020*), their locomotion is thus rather better characterized as a form of *grounded running* (*Formica polyctena*: *Reinhardt & Blickhan, 2014*).

No differences were observed in the mass specific external mechanical work nor in the mass specific external mechanical power between individuals of different sizes. This is in agreement with the literature, which shows that the mass specific external mechanical work is constant over a wide range of animal species ranging from 10g to 100 kg in body mass (*Full & Tu, 1991*; *Alexander, 2005*). The value we found in *M. barbarus* workers (mean $\pm$ SD: $1.082 \pm 0.175$ J m$^{-1}$ kg$^{-1}$) is very close to that reported in the literature for a wide variety of organisms, i.e., just above 1 J m$^{-1}$ kg$^{-1}$.

## Loaded ants

Independent of ant mass, we did not observe any changes in the mean CoM $Z$ position and in the amplitude of the oscillations of the CoM $Z$ position in loaded ants. Even if the CoM mean speed decreased in loaded ants (*Merienne et al., 2020*), this decrease seems to have little impact on the sinus-like variation of the CoM speed (Figs. 3D and 3F). On the other hand, the pattern of variation of the CoM $Z$ position was strongly affected by heavy loads. The locomotion was much more jerky and the variations in the CoM $Z$ position could not be approximated by a sinus function, especially for big ants (Fig. 3E). Moreover, because of the decrease in locomotory speed due to carrying a load (*Merienne et al., 2020*) and the amplitude of the CoM $Z$ position which remained unchanged, the amplitude of

Merienne et al. (2021), *PeerJ*, DOI 10.7717/peerj.10664

**Table 2** **Effect of body mass and load ratio on the changes in kinematics between unloaded and loaded locomotion.** The results of a power law model describing the influence of ant mass M (in mg) and load ratio LR on the relative changes of variables between the loaded and unloaded condition are indicated on each line of the table. The equation of the model is $Y_l/Y_u = cM^d LR^e$ with $Y_l$ and $Y_u$ the value of the variable in the unloaded and loaded condition, respectively. The first column gives the model prediction, along with its 95% confidence interval for the mean value of ant masses (12.5 mg) and a load ratio of 1 (unloaded ants). The second, third and fourth column give the value of the coefficients c and d for ant mass, and that of the coefficient e for load ratio, respectively, along with their 95% confidence interval. The adjusted R for the model is given in the fifth column. If the value of a coefficient is positive (i.e. c, d or e) this means that the value of Y in loaded condition increases compared to unloaded condition when the explanatory variable increases and vice versa. Bold characters indicate that 0 is not included in the 95% confidence interval of the coefficient d for ant mass and e for load ratio. Because ants moved along a straight path, we averaged the values of the variables for the right and left leg of each pair of legs. $N = 52$ ants.

| | Variable (ratio loaded / unloaded) | Model prediction for mean (ant mass) and LR=1 [CI] | Coefficient c [CI] | Coefficient for d for ant mass [CI] | Coefficient for e for load ratio [CI] | Adj $R^2$ |
|---|---|---|---|---|---|---|
| 1 | RMSE Speed norm | 0.912 [0.679;1.225] | 0.700 [0.413;1.187] | 0.104 [−0.033; 0.241] | 0.208 [−0.062; 0.478] | 0.02 |
| 2 | RMSE Z position | 0.863 [0.615;1.212] | 0.584 [0.318;1.072] | 0.154 [−0.004; 0.311] | **0.412 [ 0.101; 0.722]** | 0.10 |
| 3 | Z position amplitude (BL[1]) | 1.242 [0.836;1.845] | 1.115 [0.548;2.266] | 0.042 [−0.141; 0.226] | 0.011 [−0.352; 0.373] | 0.02 |
| 4 | Mean Z position (BL) | 0.917 [0.755;1.113] | 0.874 [0.617;1.238] | 0.019 [−0.071; 0.109] | −0.062 [−0.240; 0.116] | 0.01 |
| 5 | Body angle (°) | 0.884 [0.440;1.774] | 0.645 [0.175;2.378] | 0.120 [−0.226; 0.467] | −0.622 [−1.274; 0.030] | 0.09 |
| 6 | Correlation coefficient | 1.353 [0.835;2.194] | 0.996 [0.419;2.366] | 0.121 [−0.104; 0.345] | −0.212 [−0.654; 0.230] | 0.04 |
| 7 | Percentage congruity (%) | 1.116 [1.012;1.231] | 1.186 [0.995;1.414] | −0.024 [−0.069; 0.022] | **−0.176 [−0.266;−0.086]** | 0.22 |
| 8 | Ek / Ep phase (°) | 2.174 [0.766;6.171] | 12.07 [1.816;80.30] | **−0.663 [−1.183;−0.143]** | **−0.995 [−1.988;−0.001]** | 0.14 |
| 9 | Mass specific Wext (nJ/mm/mg) | 1.120 [0.917;1.367] | 0.852 [0.596;1.218] | **0.107 [ 0.015; 0.200]** | **0.454 [ 0.271; 0.636]** | 0.31 |
| 10 | Mas specific Pext (nJ/s/mg) | 1.202 [0.862;1.676] | 1.153 [0.636;2.091] | 0.016 [−0.138; 0.171] | −0.255 [−0.559;0.049] | 0.04 |
| 11 | Percentage recovery (%) | 0.883 [0.571;1.367] | 1.090 [0.498;2.384] | −0.082 [−0.285;0.120] | −0.144 [−0.544;0.255] | 0.01 |

**Notes.**
BL, Body Length

the variation of the CoM potential energy became much greater than that of the kinetic energy (Figs. 4C–4F). The mechanical energy required to raise the CoM in loaded ants is thus much greater than that required to accelerate it in the forward direction. Therefore, the variations in the CoM potential energy and in the CoM mechanical energy are nearly identical and the external mechanical work is mostly achieved for raising the CoM.

Independent of ant mass, the mass specific mechanical work increased with load ratio. This is an unexpected result as the mass specific mechanical work is independent of load ratio in humans (*Bastien et al., 2016*). It is thus mechanically more costly for ants to move one unit of mass on one unit of distance during loaded locomotion than during unloaded locomotion. Moreover, independent of load ratio, the mass specific mechanical work increased with ant mass, which means that the mechanical work big ants have to perform in order to raise one unit mass of their body on one unit of distance is greater than that of small ants.

Compared to unloaded locomotion, none of the gait parameters we studied was modified in a discrete way in loaded locomotion. We conclude that ants do not use a specific gait in order to carry a load. Rather, they adapt their locomotion to the mass of the load they transport.

In this study we focused only on the external mechanical work ants have to perform in order to raise and accelerate their CoM. Therefore, we did not take into account the movement of the leg segments in the determination of both the position of the overall CoM and the internal mechanical work that ants have to perform in order to accelerate their legs relative to their CoM. *Kram, Wong & Full (1997)* found in the cockroach *Blaberus discoidalis* that this internal work represents about 13% of the external mechanical work generated to lift and accelerate the CoM. Considering that the stride frequency of *M. barbarus* (mean ± SD: 4.8 ± 0.9 Hz, *Merienne et al., 2020*) is lower than that of *B. discoidalis* (mean ± SD: 6.8 ± 0.8 Hz, *Kram, Wong & Full, 1997*), if one assumes that the mass of the legs of *M. barbarus* workers represents the same percentage of total body mass as that of *B. discoidalis*, i.e., 10–12% (*Kram, Wong & Full, 1997*), we would expect the internal mechanical work to represent a smaller part of the total mechanical work in *M. barbarus* compared to *B. discoidalis*. Despite the technical difficulties for tracking the 3D displacement of insect legs (but see: *Uhlmann et al., 2017*), this aspect could constitute an interesting perspective for further studies.

## CONCLUSION

Unloaded ants adopted different postures according to their size. Small ants were more erected on their legs than big ants and their CoM showed greater vertical oscillations. However, this did not affect the amount of energy per unit of distance and unit of body mass required to raise and accelerate their CoM. Both for unloaded and loaded locomotion, the kinetic and potential energies were mainly in phase, which corresponds to the grounded-running gait described by *Reinhardt & Blickhan (2014)* during unloaded locomotion in the ant *Formica polyctena*. Regarding loaded locomotion, the amount of energy needed to raise and accelerate the center of mass per unit of distance and unit

of body mass increased with increasing body mass and load mass, suggesting that, in this respect, smaller ants carrying smaller loads were mechanically more efficient during locomotion. This could be related to the division of labor observed on the foraging trails of *M. barbarus*. In fact, relative to the proportion they represent on foraging trails, workers of intermediate size, i.e., *media*, contribute the largest share of seed transport, compared to small or big workers. Big workers are mostly present at the end of the trails where they climb on the plants to cut thick stalks or spikelets, or inside the nest, to mill the seeds and prepare them for consumption.

## ACKNOWLEDGEMENTS

The authors wish to thank Ewen Powie and Loreen Rupprecht for their help in video analysis and data extraction. Thanks are also due to Melanie Debelgarric for designing the Dufour gland extraction protocol.

### Funding

Hugo Merienne was funded by a doctoral grant from the French Ministry of Higher Education, Research and Innovation through the SEVAB graduate school of the University of Toulouse. The Image acquisition equipment was financed by the project Serious GaRS (ref No 16004115/MP0007086) funded by FEDER-FSE Midi-Pyrénées 2014–2020. The funders had no role in study design, data collection and analysis, decision to publish, or preparation of the manuscript.

### Grant Disclosures

The following grant information was disclosed by the authors:
French Ministry of Higher Education.
FEDER-FSE Midi-Pyrénées 2014–2020.

### Competing Interests

The authors declare there are no competing interests.

### Author Contributions

- Hugo Merienne conceived and designed the experiments, performed the experiments, analyzed the data, prepared figures and/or tables, authored or reviewed drafts of the paper, and approved the final draft.
- Gérard Latil performed the experiments, prepared figures and/or tables, ant collection and colony maintenance, and approved the final draft.
- Pierre Moretto conceived and designed the experiments, authored or reviewed drafts of the paper, project administration and funding, and approved the final draft.
- Vincent Fourcassié conceived and designed the experiments, analyzed the data, prepared figures and/or tables, authored or reviewed drafts of the paper, and approved the final draft.

## Data Availability

Raw data are available as a Supplementary File.

## Supplemental Information

Supplemental information for this article can be found online at http://dx.doi.org/10.7717/peerj.10664#supplemental-information.

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
