# Peer review of "Dynamics of locomotion in the seed harvesting ant Messor barbarus: effect of individual body mass and transported load mass"

_PeerJ, doi:10.7717/peerj.10664_

## Round 0.1 · original submission · Minor Revisions

As you'll see below, both reviewers were enthusiastic about your manuscript. However, Reviewer 2 identified a number of small issues that need to be addressed before the manuscript could be accepted.

Reviewer 1 ·

Basic reporting

The manuscript is well structured, clearly written and uses a well comprehendable language. The literature references both in the introduction and discussion comprehensively cover the field. The manuscript is very well structured and follows a clear logic. The raw data is present and in an impeccable state. The overall topic is self-contained and follows a clear question throughout.

Experimental design

The expriment is original and fits within the scop of the journal. The research question is well defined and relevant within the field of animal locomotion. It fills a knowledge gab.
The invesitation is appropriate and of high standard. The methods used are sufficiently described.

Validity of the findings

The findings are novel and hence add important knowledge to the field.
The data are provided and to my judgment the analsys is sound. The conclusions follow the data and are not over-interpreting.

Additional comments

This is a very well written manuscript, however with a one or the other typo, i.e. line 152 United Sates. Please check prior to publishing.

Reviewer 2 ·

Basic reporting

I noticed some typos and grammar errors.

The figure panels are explained in the figure legends, but adding color legends and titles (e.g. no load, small load, high load in figures 3 and 4) directly to all figures would greatly improve readability.

Experimental design

The research question was not clear to me. In lines 98-103, the authors identify the knowledge gap. But what is the specific hypothesis being tested? What is the expected effect of body mass and added load? Would differently sized/loaded animals be expected to have a different movement strategy or vary with respect to efficiency? A clear hypothesis would help streamline the manuscript (much like in Merienne et al. 2020).

The Methods section lacks some details.

Were the data in this manuscript collected as part of Merienne et al. (2020)? If so, this should be noted in the Methods section.

Please add information about the weight of the added load in the manuscript (line 151). Based on the datasheet, it looks like the load varied from ant to ant. Why was that?
Did the loads significantly affect the overall COM for each ant? By how much?

How many strides per animal were analyzed (lines 182-184)?

How are sinusoids fitted to Ek and Ep in the high-load condition, in which oscillations do not look sinusoidal (line 228 and Figure 4E,F)?

What was the rationale for expressing variables as a power law function of ant mass (lines 223-234)?

At the end of the Methods section, it would be helpful to add information on how the CIs were calculated and how they were used to determine the statistical significance of the model coefficients.

Validity of the findings

As stated above, it would help to have a clearly defined research question/hypothesis to which the conclusions can be linked to.

Additional comments

This manuscript is an interesting extension to the authors’ prior study on load carrying in ants and its effects on locomotion. I have no serious concerns about the conclusions drawn, but I have a number of additional comments (in order of appearance) that I feel should be addressed to improve the readability of the manuscript and the repeatability of the experiments.

The introduction feels very gait-centered. Based on the first paragraphs, I would have expected to see a more thorough gait analysis including leg kinematics. It would be helpful if the introduction reflected the focus of the manuscript (load-dependent movements of the COM rather than gaits) more clearly. For example, instead of introducing the concept of gaits and COM dynamics in line 46, the authors could first emphasize that insects show remarkable locomotor flexibility dependent on task/context, but that the effect of load, specifically the mechanical energy transfer, is not well understood. Then, they could introduce Messor barbarous as a model system, measuring the COM dynamics as a way of quantifying the effects of load carriage, and the specific research question(s).

How fast were the ants walking in the present study? Do the COM dynamics (amplitudes, phase shifts, …) depend on walking speed?

Line 265: “…Ek and Ep were more in phase for big ants than small ants”. Isn’t it the other way around according to Figure 5A and Table 1?

Figure 1: Why are there four/five tracked points in A,C but six points in B,D? Were the points tracked in all five camera views? Also, it would help to indicate the (overall) COM position and highlight the added load (maybe with an arrow?).

Figure 2: I don’t count 52 data points. Are data missing or were data excluded? Also, why was the distinction between small and big ants made at 10.2 mg? Is this the median weight?

Figure 5: The Welch-test was used to test small ants vs big ants. Was it also used to test unloaded ants vs loaded ants? This is what the letters “a” and “b” suggest to me (e.g., in panel A, small unloaded and small loaded ants do not differ signficitanly; they both belong to group “a”). If so, a paired test should be used instead. If not, it might be clearer to draw a bar above the experimental groups that were compared with each other rather than using letters.

Figure 6: Why is the mechanical work the only parameter that is not shown for the loaded ants?

---

## Round 0.2 · accepted · Accept

I believe you properly addressed all of the (minor) issues raised by the reviewers.